# LC–MS^3^ Strategy for Quantification of Carbamazepine in Human Plasma and Its Application in Therapeutic Drug Monitoring

**DOI:** 10.3390/molecules27041224

**Published:** 2022-02-11

**Authors:** Dongxiao Ma, Zhengchao Ji, Haiwei Cao, Jing Huang, Lei Zeng, Lei Yin

**Affiliations:** 1Department of Clinical Laboratory, The First Hospital of Jilin University, Jilin University, Changchun 130021, China; madx@jlu.edu.cn (D.M.); jizc@jlu.edu.cn (Z.J.); caohwjdyy@jlu.edu.cn (H.C.); 2The Bethune Institute of Epigenetic Medicine, The First Hospital of Jilin University, Jilin University, Changchun 130021, China; 3School of Life and Pharmaceutical Sciences, Dalian University of Technology, Panjin 124221, China

**Keywords:** LC-MS^3^, carbamazepine, therapeutic drug monitoring

## Abstract

This study developed a detection method based on the strategy of HPLC/MS^3^ and verified its suitability by quantifying carbamazepine in human plasma. The high-performance liquid chromatography–tandem mass spectrometry (HPLC/MS^3^) system was performed using a Shimadzu UFLC XR liquid chromatography and a SCIEX QTRAP^®^ 5500 linear ion trap triple quadrupole mass spectrometer. The specific operation was as follows: the sample protein was firstly precipitated using methanol, then carbamazepine and carbamazepine-D_2_N^15^ were separated on an ACQUITY UPLC HSS T3 column using the gradient elution with solvent A (0.1% formic acid) and solvent B (0.1% formic acid in acetonitrile) at a flow rate of 0.25 mL/min. Each sample was run for 7 min. This method was validated for various parameters including accuracy, precision, selectivity, linearity, LLOQ, etc. Only 5 μL of sample plasma could obtain the result of LLOD 0.5 µg/mL. The intra-day and inter-day precision was <8.23%, and accuracy was between −1.74% and 2.92%. This method was successfully used for monitoring the blood concentration of epilepsy patients after carbamazepine treatment.

## 1. Introduction

Carbamazepine (C_15_H_12_N_2_O, MW: 236.27 g/mol) (shown in Figure 1) is an aromatic medicine used for anticonvulsants and analgesia, also known as Tegretol or carbamazepine [1,2,3,4,5]. Approved by the FDA in 1963, carbamazepine has been commonly used in the treatment of epilepsy and trigeminal neuralgia [6,7]. In clinical applications, carbamazepine has been found to cause many severe adverse reactions—especially in immunity, hematology, skin, kidney, and liver [8,9,10]—which are associated with blood drug concentration and clinical application dosage. Therefore, therapeutic drug monitoring of carbamazepine is important in any clinical setting. It is necessary to adjust dosages to achieve the best therapeutic effect while avoiding low (subtherapeutic) and high (toxic) levels. The optimal plasma level of carbamazepine for the treatment of epilepsy is reported to be 4–12 μg/mL [11,12,13,14].

Therapeutic drug monitoring has been performed using immunoassays for a long time [15,16,17,18,19,20]. However, the selectivity of immunoassays is limited. They can suffer from non-specific interference from the biological matrix, structure similar compounds, or metabolites. Compared with immunoassays, the LC-MS/MS methods show more advantages in selectivity and sensitivity. Furthermore, LC-MS/MS methods are more accurate and precise. These factors are why LC-MS/MS methods are suggested as the “gold standard” for the determination of compounds in biological samples. Thus far, current studies have shown that carbamazepine can be quantitatively detected by several LC-MRM methods [21,22,23,24,25]. However, as far as we know, the strategy of MS^3^ detection has not yet been researched and considered in determining carbamazepine in biological samples, which is possible with Qtrap tandem mass spectrometers [26,27,28,29,30,31,32,33,34,35]. The excitation efficiency of MS^3^ is high, and the scanning rate of MS^3^ is fast. The scanning speed of MS^3^ can be up to 20,000 Da/s. The workflow of MS^3^ scanning mode is as follows: Firstly, the analyte precursor ions are selected in Q1, and then in the collision cell (Q2) the precursor ions are fragmented by collision-induced dissociation in order to generate product ions, which are then captured by the linear ion trap. Subsequently, the specific product ions are selected in the linear ion trap for secondary fragmentation; the second-generation product ions can be captured by the detector. Therefore, the MS^3^ scanning mode can achieve a more accurate quantification by performing MRM^3^ transitions. It is a detection mode with high sensitivity, selectivity, and a better detection limit.

This study aimed to develop an LC-MS^3^ method with high selectivity for the quantitative detection of carbamazepine in human plasma based on a simple and fast sample preparation. Moreover, another LC-MRM-based method was compared with this LC-MS^3^-based method. The developed LC-MS^3^ method was tested with patient plasma samples. This method only used 5 μL plasma and it provides high selectivity with excellent accuracy and precision. Based on current research reports, the validated LC-MS^3^ method is the first assay to be used to quantitatively detect carbamazepine in human plasma.

## 2. Results

### Assay Validation

For this LC-MS^3^ method, the representative chromatograms of carbamazepine and carbamazepine-D_2_N^15^ are shown in Figure 2 and suggest that the retention time of both was hardly affected by endogenous substances in human plasma. In addition, for the blank plasma sample, the response enhancement of carbamazepine and carbamazepine-D_2_N^15^ was not observed, which suggested that the carry-over was negligible and there was no crosstalk between MS channels. The linear range for the determination of carbamazepine was 0.50–30 µg/mL, and the typical regression equation was y = 17.4x + 1.52 (r = 0.9973). For all concentrations, the precision was <2.92%, the intra- and inter-day accuracy was <8.23%. As shown in Table 1, the recovery of the carbamazepine-D_2_N^15^ was 98.9–110.2%. The recovery rate of carbamazepine is reproducible within the concentration range studied.

For the matrix effect, the actual concentration as a percentage of the nominal concentration for the three-level QC samples were: 94.3 ± 8.66% (low), 107.7 ± 6.83% (medium), 112.5 ± 4.11% (high).

The stability data of carbamazepine is shown in Table 2. The concentration of carbamazepine was maintained at ±7.23% of the nominal concentration under all test conditions, indicating that carbamazepine is essentially stable.

## 3. Discussion

### 3.1. Optimization of MS Conditions

The MS^2^ and MS^3^ spectra of carbamazepine and carbamazepine-D_2_N^15^ are shown in Figure 3. Since they are all nitrogen-containing chemicals, they react well in the positive ionization mode. For the medium-resolution MRM acquisition, quantitating for carbamazepine used a transition of *m/z* 237.0→220.1; for carbamazepine-D_2_N^15^, a transition of *m/z* 240.0→196.2 was used (Figure 3). In the MS^3^ mode, for carbamazepine the product ions at *m/z* 220.1 are fragmented to the second-generation product ions at *m/z* 192.2, of which 165.1 then undergo MS^3^ transitions. While for carbamazepine-D_2_N^15^, the product ions at *m/z* 196.2 are fragmented to the second-generation product ions at *m/z* 181.2 to 167.2. The selected transitions were as follows: For carbamazepine, *m/z* 237.0→220.1→192.2; for carbamazepine-D_2_N^15^, *m/z* 240.1→196.2→181.2. For second-generation product ions, the mass range scanned was ±1.0 Da.

### 3.2. Optimization of LC Conditions

An ACQUITY UPLC HSS T3 column (2.1 × 100 mm, 1.8 µm) was selected for chromatography because it provides good peak shapes and retention behaviors of carbamazepine and carbamazepine-D_2_N^15^. The use of high-organic phase gradient elution could also effectively reduce residues and have a positive effect on eliminating matrix effects. The retention time of both was 3.25 min with optimal LC conditions.

### 3.3. Optimization of Sample Processing

In this study, methanol precipitation of proteins was selected for sample processing, which has the advantages of convenience and simplicity. Internal standard solution (5 µL) and methanol (1000 µL) were added to 5 µL of plasma to precipitate proteins. It was found that the protein precipitation sample that was diluted 202 times showed symmetrical peaks with better sensitivity and the data could not be affected by the matrix effect of carbamazepine or carbamazepine-D_2_N^15^. Therefore, the methanol-precipitated protein was applied in this study.

### 3.4. Comparison of LC–MS^3^ and LC-MRM Methods

Compared with the LC-MS^3^ method, the LC-MRM method, using MRM transitions at *m/z* 237.0→220.1 for carbamazepine and *m/z* 240.1→196.2 for carbamazepine-D_2_N^15^, was optimized. The chromatogram of 0.50 μg/mL carbamazepine was acquired using the LC-MRM method (Figure 4BI) and the LC-MS^3^ method (Figure 2BI). The peak height of carbamazepine at 0.50 μg/mL for MRM acquisition is 1449.5 cps and S/N is 19.6, while the peak height for MS^3^ acquisition is 1.2 × 10^6^ cps and S/N is 60.5. MS^3^ scan exhibits a higher level of sensitivity. Carbamazepine’s S/N was significantly improved in the additional fragmentation step.

A total of 34 patients who were treated with carbamazepine (100–1100 mg/day) were monitored for blood drug concentration with the validated LC-MS^3^ method. The concentrations of carbamazepine detected by the two methods are shown in Table 3 and Figure 5. Comparison of the plasma concentration result measurements with the estimated values is shown in Figure 6 and Figure 7. The Passing–Bablok method was used for regression analysis, and the results showed that the data had a strong consistency with no constant deviation and proportional deviation (shown in Figure 6), where y = −0.105109 (95% CI, −0.3767 to 0.1084) + 0.934783 (95% CI, 0.9059 to 0.9757)x. A Bland–Altman plot showed the difference between the LC-MRM and LC-MS^3^ methods was −7.7% (95% LoA, −17.9–2.6%). As shown in Figure 7, carbamazepine differences are evenly distributed on both sides of the mean, except for the value of 2/34 (5.88%); among them, the maximum concentration deviation corresponding to 94.1% of carbamazepine samples is ±1.96 SD. Therefore, LC-MS^3^ can perform carbamazepine drug monitoring through calculation of carbamazepine concentration, similar to LC-MRM.

## 4. Materials and Methods

### 4.1. Reagents 

Carbamazepine was supplied by the National Institute for the Control of Pharmaceutical and Biological Products, Beijing, China. Internal standard carbamazepine-D_2_N^15^ was provided by Chemsky International Co., Shanghai, China (Figure 1). Acetonitrile was supplied by Fisher Scientific (Fair Lawn, NJ, USA). The distillated water used in the entire study was prepared from demineralized water, and all chemical reagents were HPLC grade.

### 4.2. LC–MS^3^ Conditions

The chromatographic system consisted of the Shimadzu UFLC XR system (Shimadzu, Kyoto City Japan), which had a 40 °C column oven, an automatic sampler at 4 °C, and two binary pumps. Then, the ACQUITY UPLC HSS T3 column (2.1 × 100 mm, 1.8 µm) was used for the separation of carbamazepine and carbamazepine-D_2_N^15^; the gradient elution with solvent A (0.1% formic acid) and solvent B (0.1% formic acid in acetonitrile) was adjusted to 0.25 mL/min. The elution gradient was: 0–2 min, 50% to 50% B; 2–3 min, 50% to 65% B; 3–3.5 min, 65% to 75% B; 3.5–4 min, 75% to 90% B; 4–4.5 min, 90% to 90% B; 4.5–4.6 min, 90% to 50% B; and 4.6–7 min, 50% to 50% B.

The positive-ion-mode electrospray ionization Q-Trap 5500 (Sciex, Concord, ON, Canada) was used for tandem mass spectrometry. In MS^3^ mode, the product ions could be split in a linear trap under certain conditions into second-generation product ions, which then went through the MS^3^ transition. The transition was as follows: carbamazepine, *m/z* 237.0→220.0→192.0; IS, *m/z* 240.1→196.2→181.0. For second-generation product ions of two homologous compounds, the mass range scanned was ±1.0 Da. The conditions of MS could be improved by injecting carbamazepine and IS standard solution (10 µg/mL).

### 4.3. Calibration Standards and Preparing QC Samples 

Carbamazepine in methanol solution was used as a stock solution (1.00 mg/mL). Then, the blank plasma was added into the stock solution to produce final concentrations of 0.50, 1.00, 2.00, 5.00, 10.0, 15.0, and 30.0 μg/mL as calibration standards. By the same method, three QC samples (1.00 μg/mL, 5.00 μg/mL, and 15.0 μg/mL) were prepared. The internal standard solution was prepared with carbamazepine-D_2_N^15^ in methanol and diluted to 10.0 μg/mL with methanol aqueous solution with a ratio of methanol to water was 50:50 (*v/v*).

### 4.4. Sample Processing

Frozen plasma samples were thawed in a room-temperature water bath kettle.

A total of 5 μL plasma was added to 5μL IS working solution and 1000 μL methanol, then the test tube was vortexed for 1 min, and centrifuged at 12,000 rpm for 5 min at 4 °C. A total of 2 μL volume of the supernatant was injected into the LC-MS system.

### 4.5. Assay Validation

According to the U.S. Food and Drug Administration’s bio-analytical method validation system, the LC-MS^3^ strategy was validated. Selectivity was tested using six different blank human plasma samples, the concentration–response relationship, as well as an appropriate weighting scheme and regression equation. Based on analyte peak area ratios, the linearity was tested by linear least-squares regression with 1/x^2^ as standard curves’ weighting index. Two copies of IS were prepared on 3 different days. The precision for intra- and inter-day (R.S.D.) was validated by six QC samples in order to replicate the experiment on 3 continuous days. Accuracy (R.E.). LLOQ was determined with accuracy ±20% and precision <15%. Matrix effects were validated by comparing peak areas of analytes and IS in standard solution with those in spikes samples post extracting. Recovery was determined by comparing peak areas of QC samples with those of the post-extracting blank plasma spiked at certain concentrations. QC samples were stored for 14 days at −70 °C and thawed for 1 h at 25 °C, which was one freeze/thaw cycle. After three freeze/thaw cycles, stability was validated. Analyte’s stability in samples stored in the auto-sampler vials at 4 °C for 6 h was also tested. 

### 4.6. Blood Concentration Monitoring

To investigate the suitability of this analysis method, patients with epilepsy who received carbamazepine intervention were selected as the subjects of this study and their plasma samples were analyzed to quantify the level of carbamazepine in plasma. Blood samples were collected from patients before taking carbamazepine every morning. This study was approved by an independent ethics committee. All subjects signed an informed consent form. The clinical applicability of this method is evaluated by quantifying the carbamazepine in the plasma sample of patients, and the reference value comes from the routine carbamazepine blood concentration monitoring of the hospital’s therapeutic drug monitoring laboratory.

### 4.7. Statistical Analysis

Analysis1.6.3 software (Sciex, Concord, ON, Canada), Microsoft 2010 (Microsoft, Redmond, WA, USA), and MedCalc Version 15.2.2 (MedCalc Software Ltd, Beijing, China) were adopted for data acquisition, data processing, and graphic presentation. Bland–Altman analysis and Passing–Bablok regression were used to evaluate the consistency between the concentration of carbamazepine detected from MRM and MS^3^. The method was considered to be suitable if the differences were within 1.96 SD of the mean difference for ≥67% of the sample pairs. Carbamazepine plasma concentration was calculated based on MRM and MS^3^ data, respectively.

## 5. Conclusions

The LC-MS^3^ method developed by this research institute has the advantages of high sensitivity, high signal-to-noise ratio, and small sample size (5 µL) requirement. The verification of the resulting data shows that the concentration of carbamazepine in human plasma can be quantified in order to achieve effective clinical blood concentration monitoring. In this study, for the first time, LC-MS^3^ was used for the quantitative detection of carbamazepine concentration in human plasma, which provided a basis and evidence for the potential application of LC-MS^3^ for the detection of compounds in biological samples.

## Figures and Tables

**Figure 1 molecules-27-01224-f001:**
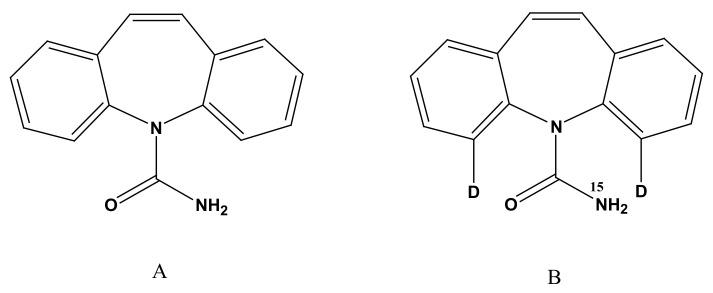
Chemical structure of carbamazepine (**A**) and carbamazepine-D_2_N^15^ (**B**).

**Figure 2 molecules-27-01224-f002:**
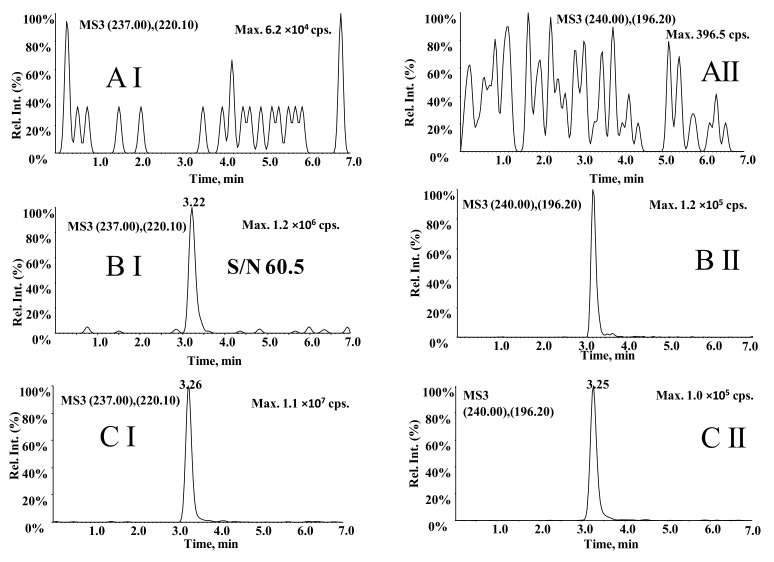
Representative LC-MS^3^ chromatograms of carbamazepine (**I**) and carbamazepine-D_2_N^15^ (**II**). (**A**) blank plasma, (**B**) blank plasma added to carbamazepine at 0.50 μg/mL of LLOQ and 10.0 μg/mL of IS, and (**C**) a blood sample from an epilepsy patient after oral administration of carbamazepine.

**Figure 3 molecules-27-01224-f003:**
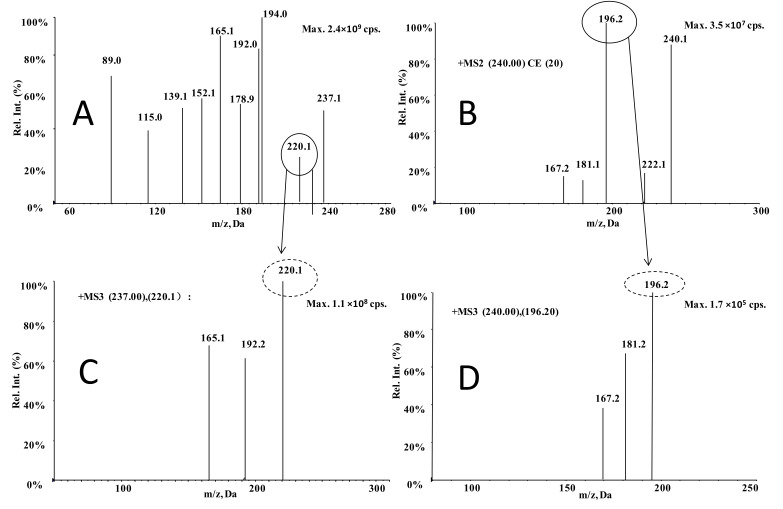
Production and MS^3^ spectra of carbamazepine (**A**,**C**) and carbamazepine-D_2_N^15^ (**B**,**D**).

**Figure 4 molecules-27-01224-f004:**
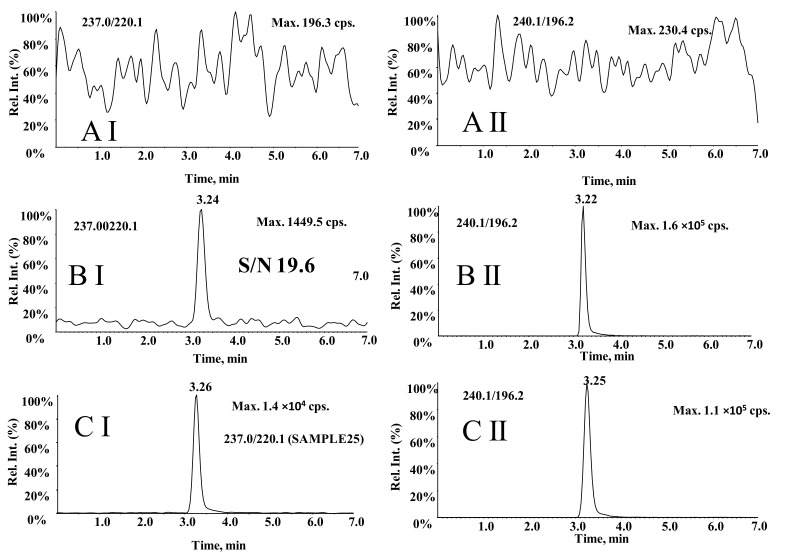
Representative LC-MRM chromatograms for carbamazepine (**I**) and carbamazepine-D_2_N^15^ (**II**). (**A**) blank plasma, (**B**) blank plasma added to carbamazepine at 0.50 μg/mL of LLOQ and 10.0 μg/mL of IS, and (**C**) a blood sample from an epilepsy patient after oral administration of carbamazepine.

**Figure 5 molecules-27-01224-f005:**
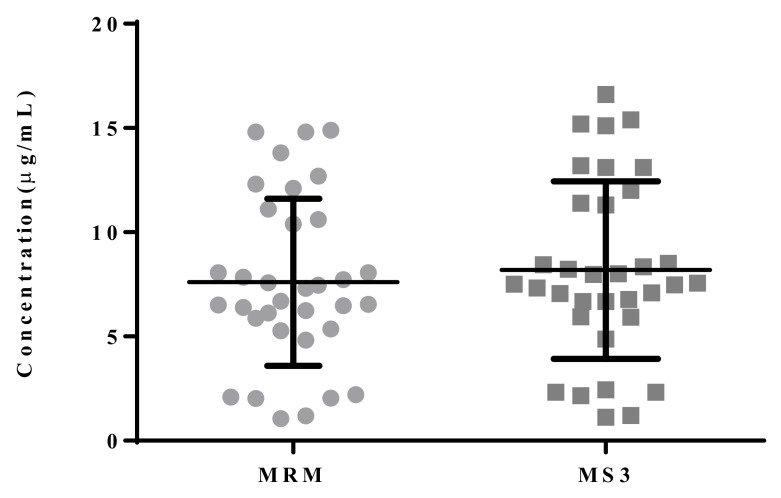
Quantitative analysis of carbamazepine for 34 patient samples comparing LC-MS^3^ and LC-MRM.

**Figure 6 molecules-27-01224-f006:**
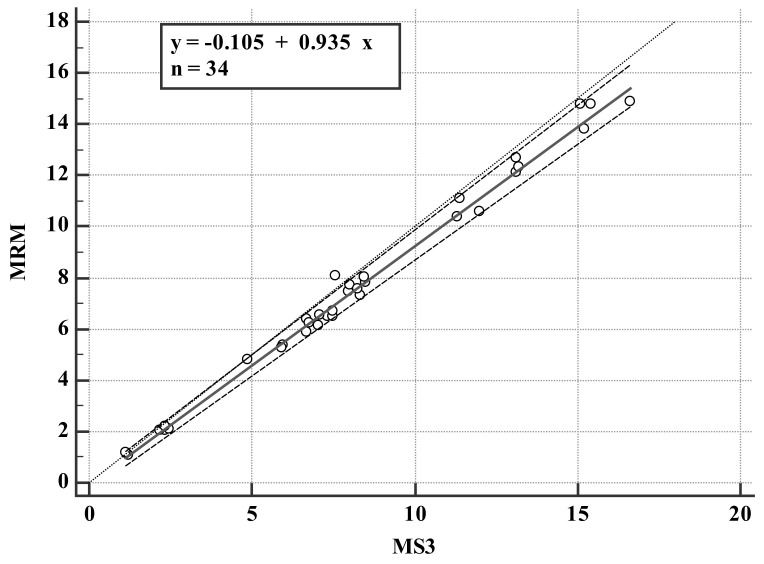
Comparison of carbamazepine concentration of patient samples measured by LC-MRM and LC-MS^3^. The solid black lines is the Passing–Bablok regression.

**Figure 7 molecules-27-01224-f007:**
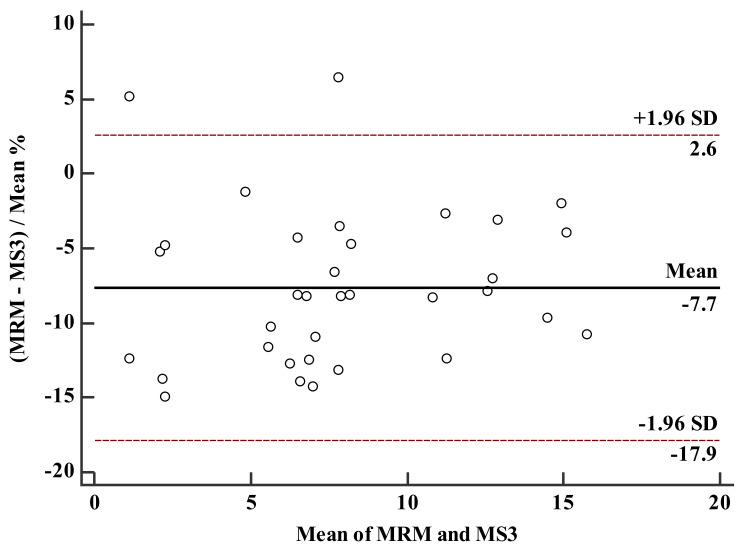
Bland–Altman plots for carbamazepine measurements illustrating the differences between MRM and MS^3^. The dots between the dotted lines indicate that the sample pairs are within the acceptable range of ±1.96 SD.

**Table 1 molecules-27-01224-t001:** Absolute recovery of carbamazepine in plasma. (Data is mean ± SD, *n* = 3).

	Recovery (%)		
	Low QC	Medium QC	High QC
Carbamazepine	100.7 ± 9.20	110.5 ± 7.00	110.2 ± 4.00

**Table 2 molecules-27-01224-t002:** Stability for carbamazepine (Data are mean ± SD, *n* = 3).

Compound	Nominal Conc. (μg/mL)	Long Term −70 °C	Short Term	Freeze– Thaw	Post- Preparative
Carbamazepine	1.0	99.9 ± 3.77	105.0 ± 2.65	98.0± 5.75	101.5 ± 7.21
	5.0	96.4 ± 1.51	101.5 ± 7.23	100.4 ± 5.25	91.6 ± 3.52
	15	94.4 ± 2.52	98.4 ±3.67	94.9 ± 3.01	89.3 ± 0.67

**Table 3 molecules-27-01224-t003:** Carbamazepine drug concentration of 34 patients quantified by LC-MRM and LC-MS^3^.

Sample ID	MRM	MS^3^	%
sample1 (74 years, female)	6.39	6.67	95.8
sample2 (34 years, female)	8.06	7.56	106.6
sample3 (40 years, male)	2.03	2.33	87.1
sample4 (29 years, male)	1.06	1.2	88.3
sample5 (23 years, male)	11.1	11.4	97.4
sample6 (61 years, male)	12.1	13.1	92.4
sample7 (34 years, female)	13.8	15.2	90.8
sample8 (37 years, female)	14.8	15.4	96.1
sample9 (30 years, male)	5.87	6.67	88.0
sample10 (33 years, female)	7.31	8.34	87.6
sample11 (27 years, male)	2.1	2.44	86.1
sample12 (29 years, male)	2.05	2.16	94.9
sample13 (25 years, male)	5.36	5.94	90.2
sample14 (57 years, female)	14.9	16.6	89.8
sample15 (23 years, male)	6.47	7.33	88.3
sample16 (30 years, female)	7.84	8.51	92.1
sample17 (34 years, male)	6.13	7.05	87.0
sample18 (23 years, female)	6.5	7.5	86.7
sample19 (33 years, female)	7.46	7.97	93.6
sample20 (39 years, female)	2.21	2.32	95.3
sample21 (9 years, male)	1.19	1.13	105.3
sample22 (14 years, male)	10.6	12	88.3
sample23 (58 years, female)	12.7	13.1	96.9
sample24 (20 years, female)	5.27	5.92	89.0
sample25 (40 years, female)	6.24	6.77	92.2
sample26 (36years, male)	7.72	8	96.5
sample27 (29 years, female)	10.4	11.3	92.0
sample28 (39 years, female)	12.3	13.2	93.2
sample29 (52 years, male)	14.8	15.1	98.0
sample30 (19 years, female)	6.54	7.1	92.1
sample31 (25 years, female)	8.05	8.44	95.4
sample32 (20 years, male)	6.70	7.48	89.6
sample33 (38 years, female)	7.58	8.23	92.1
sample34 (29 years, male)	4.82	4.88	98.8
mean	7.60	8.19	
SD	4.01	4.25	

## Data Availability

Data available on request due to restrictions, e.g., privacy or ethical. The data presented in this study are available on request from the corresponding author.

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
