# Peer review of "LC–MS3 Strategy for Quantification of Carbamazepine in Human Plasma and Its Application in Therapeutic Drug Monitoring"

_molecules, 2022, doi:10.3390/molecules27041224_

Round 1

Reviewer 1 Report

The manuscript is well-witten and easy to read, but at the moment unfortunately, I cannot tell if it is worth publishing or not. The Authors cite some similar papers [21-25] which are also about HPLC-MS(/MS) methods to determine carbamazepine in human plasma, but the Authors did not compare the analytical parameters (linearity, LOQ, LOD, etc) of these methods to their newly developed method. I understand that the Authors’ method use MS^3 technique and in the literature only MS^2 methods can be found, but this fact alone does not warrant publication and it does not tell anything about the comparison of analytical parameters. The Authors should clearly tell why their method is superior to the methods in the literature and should compare their result to the literature. Without this comparison the referee cannot tell the practical utility of the new method.

The Authors claim that “The peak height of carbamazepine at 0.50 μg/mL for MRM acquisition is 1449.5 cps …, while the peak height for MS3 acquisition is 1.2e6 cps ….” In theory this cannot happen, using one more tandem mass spectrometry step, the absolute intensity cannot be higher by almost 3 orders of magnitude. There must be some problem (different conditions, collision energy is not optimally chosen, excitation times are different, isolation window different, etc), the Authors should check and correct it.

I support the publication of the manuscript if the above mentioned problems are resolved.

Minor comment:

  • “The excitation efficiency of MS3 is higher and the scanning rate of MS3 is faster.” This is actually not true, the degree of excitation depends on the type of secondary excitation and the maximum scanning rate depends on the type of analyzer, it cannot be faster in MS3 mode than in MS2 mode. The Authors should check and rephrase it.
  • Abstract “tandem mass spectrometry cubed” is quite unusual term, just use “tandem mass spectrometry (MS^3)”

Author Response

  1. The manuscript is well-written and easy to read, but at the moment unfortunately, I cannot tell if it is worth publishing or not. The Authors cite some similar papers [21-25] which are also about HPLC-MS/MS methods to determine carbamazepine in human plasma, but the Authors did not compare the analytical parameters (linearity, LOQ, LOD, etc) of these methods to their newly developed method. I understand that the Authors’ method use MS3 technique and in the literature only MS2 methods can be found, but this fact alone does not warrant publication and it does not tell anything about the comparison of analytical parameters. The Authors should clearly tell why their method is superior to the methods in the literature and should compare their result to the literature. Without this comparison the referee cannot tell the practical utility of the new method.

â–² Thanks for the suggestion. The novelty of the presented methodology is the MS3 technique resulting in enhanced selectivity and sensitivity. There are several LC-MRM assays published(reference21-25). However, sample consumed volume, injection volume and sample processing of these studies are really different from our study. That is the reason why we did not compared our study with these studies. LLOQ of 0.50 µg/mL is sufficient in this study. Furthermore, the LLOQ of this assay could be easily reduced by using more plasma or less dilution or more injection volume or reducing to dryness and reuptake in LC eluent. Therefore, in our study, we compared the LC-MS3 and LC-MRM method with the same sample processing method(both the same sample consumed volume and same injection volume). MS3 scan exhibits a higher level of sensitivity. Carbamazepine’s S/N was significantly improved in the additional fragmentation step.This work is also a proof of concept for using LC-MS3 technique to determination of chemicals in biological samples.

  1. The Authors claim that “The peak height of carbamazepine at 0.50 μg/mL for MRM acquisition is 1449.5 cps …, while the peak height for MS3 acquisition is 1.2e6 cps ….” In theory this cannot happen, using one more tandem mass spectrometry step, the absolute intensity cannot be higher by almost 3 orders of magnitude. There must be some problem (different conditions, collision energy is not optimally chosen, excitation times are different, isolation window different, etc), the Authors should check and correct it. I support the publication of the manuscript if the above mentioned problems are resolved.

â–² Thanks for the question. Compared with MRM acquisition, MS3 scan could maintain a higher level of sensitivity. The novelty of the presented methodology is the MS3 technique resulting in enhanced selectivity and sensitivity. The workflows of MS3 mode and MRM mode are appended below:

Figure 1 The workflows of MS3 mode and MRM mode

In MS3 scanning mode, the analyte precursor ions are first selected in Q1 and then the precursor ions are fragmented into product ions via collision induced dissociation in collision cell (Q2), the product ions generated in Q2 are enriched firstly and then captured in Q3 (linear ion trap). Finally, the selected product ions are further fragmented in linear ion trap and the second-generation fragment ions are scanned out to the detector. MS3 technique has its own advantages because it can perfom three stage fragmentation. In addition, fragment ions can be enriched in the ion trap for a period of time before further fragmentation. Therefore, the selectivity and sensitivity of MS3 technique is significantly improved.

  In MRM scanning mode, the analyte precursor ions are first selected in Q1 and then the precursor ions are fragmented into daughter ions via collision induced dissociation in collision cell (Q2), and then daughter ions were captured in Q3.

  As showed in figure 2 and 3, the peak height of carbamazepine at 0.50 μg/mL for MRM acquisition is 1449.5 cps, the background noise is about 196 cps and signal to nosie ratio is about 19.6; While the peak height for MS3 acquisition is 1.2e6 cps, the background noise is about 6.0e4cps and signal to nosie ratio is about 60.5. The absolute intensities of analyte and noise are both increased. Finally, the singnal to nosie ration is improved about 3 times.

  Compared with MRM, the MS3 mode have many advantages such as high sensitivity, high scan rate (20000 Da/s) and so on. But, The MS3 mode also had some disadvantages such as a relative narrow linear range.

Figure 2. Representative LC-MRM chromatograms for carbamazepine(I) and carbamazepine-D2N15(II). (A) blank plasma, (B) blank plasma added with carbamazepine at the 0.50 μg/mL of LLOQ and 10.0 μg/mL of IS, (C) a blood sample from a epilepsy patient after oral administration of carbamazepine.

Figure 3. Representative LC-MS3 chromatograms of carbamazepine (I) and carbamazepine-D2N15(II). (A) blank plasma, (B) blank plasma added with carbamazepine at the 0.50 μg/mL of LLOQ and 10.0 μg/mL of IS, (C) A blood sample from a epilepsy patient after oral administration of carbamazepine.

Minor comment:

3.“The excitation efficiency of MS3 is higher and the scanning rate of MS3 is faster.” This is actually not true, the degree of excitation depends on the type of secondary excitation and the maximum scanning rate depends on the type of analyzer, it cannot be faster in MS3 mode than in MS2 mode. The Authors should check and rephrase it.

â–² Thanks for the suggestion. We have revised it.

  1. Abstract “tandem mass spectrometry cubed” is quite unusual term, just use “tandem mass spectrometry (MS3)”

â–² Thanks for the suggestion. We have revised it.

Reviewer 2 Report

In my opinion, the manuscript titled ,,LC–MS3 strategy for quantification of carbamazepine in human plasma and its application in therapeutic drug monitoring’’ can be recommended for publication in Molecules, however after major revision. 
In this study, the authors developed a novel LC–MS3 method determination of carbamazepine in human plasma. The method was applied in therapeutic drug monitoring. The developed LC- MS3 based method was compared with LC-MRM based method. The obtaining findings are interesting and may have practical implications. However, I have important remarks and recommendations as follows:
1.    A very important part of the analytical method preparation is validation. In this work, the authors provided little information about the validation process. For the new analytical method, full validation should be performed and presented in accordance with recognized method validation guidances (for example ICH guidelines). The necessary number of repetitions of experiments should be carried out so that the obtained results can be properly statistically evaluated. The procedure for assay validation should be detailed described (how were the calculated values of the individual validation parameters, how many repetitions of the experiments were made)
Additionally 
1.    The basic information about study participants treated with carbamazepine should be provided, e.g. age, gender.
2.    There are some typos in the text. The text of the manuscript should be carefully checked.

Author Response

In my opinion, the manuscript titled “LC–MS3 strategy for quantification of carbamazepine in human plasma and its application in therapeutic drug monitoring’’ can be recommended for publication in Molecules, however after major revision.

In this study, the authors developed a novel LC–MS3 method determination of carbamazepine in human plasma. The method was applied in therapeutic drug monitoring. The developed LC- MS3 based method was compared with LC-MRM based method. The obtaining findings are interesting and may have practical implications. However, I have important remarks and recommendations as follows:

  1. A very important part of the analytical method preparation is validation. In this work, the authors provided little information about the validation process. For the new analytical method, full validation should be performed and presented in accordance with recognized method validation guidances (for example ICH guidelines). The necessary number of repetitions of experiments should be carried out so that the obtained results can be properly statistically evaluated. The procedure for assay validation should be detailed described (how were the calculated values of the individual validation parameters, how many repetitions of the experiments were made)

â–² Thanks for the suggestion. We have revised it. The LC-MS3 assay was validated in accordance with the biological method valida-tion guidance of the U.S. Food and Drug Administration. Selectivity was tested using 6 different blank human plasma samples. Linearity was tested by linear least-squares re-gression with a weighting index of 1/x2 of standard curves based on peak area ratios of analyte : IS prepared in duplicate on three separate days. Accuracy (as relative error (R.E.)) and intra- and inter-day precision (as relative standard deviation (R.S.D.) were based on assay of six replicate QC samples on three different days. The lower limit of quantitation (LLOQ) was defined as the lowest concentration that could be determined with accuracy±20% and precision < 15%. Matrix effects were evaluated by comparing peak areas of analytes and IS in post-extraction spiked samples with those in standard solutions. Recovery was determined by comparing peak areas of QC samples with those of post-extraction blank plasma spiked at corresponding concentrations. Stability of analytes in human plasma was evaluated in QC samples placed on storage for 14 days at -70°C, for 1 h at room temperature (25°C) and after three freeze/thaw cycles. Stability of analytes in processed samples on storage in autosampler vials at 4°C for 6 h was also tested.

Additionally

  1. The basic information about study participants treated with carbamazepine should be provided, e.g. age, gender.

â–² Thanks for the suggestion. We have revised it.

  1. There are some typos in the text. The text of the manuscript should be carefully checked.

â–² Thanks for the suggestion. We have revised the manuscript.

Round 2

Reviewer 1 Report

I accept the answers and support the publication of the manuscript.

Reviewer 2 Report

The authors addressed the issues found in the first draft. The manuscript can be published in present form.